# Anti-SARS-CoV-2 Antibodies Level and COVID-19 Vaccine Boosters among Healthcare Workers with the Highest SARS-CoV-2 Infection Risk—Follow Up Study

**DOI:** 10.3390/vaccines12050475

**Published:** 2024-04-29

**Authors:** Dagny Lorent, Rafał Nowak, Magdalena Figlerowicz, Luiza Handschuh, Paweł Zmora

**Affiliations:** 1Institute of Bioorganic Chemistry Polish Academy of Sciences, 61-704 Poznan, Poland; dlorent@ibch.poznan.pl (D.L.); rnowak@ibch.poznan.pl (R.N.); luizahan@ibch.poznan.pl (L.H.); 2Department of Infectious Diseases and Child Neurology, Poznan University of Medical Sciences, 61-701 Poznan, Poland; mfiglerowicz@ump.edu.pl

**Keywords:** vaccination, healthcare workers, hesitancy, booster dose, COVID-19, Poland

## Abstract

During the COVID-19 pandemic, several vaccines were developed to limit the spread of severe acute respiratory syndrome coronavirus 2 (SARS-CoV-2). However, due to SARS-CoV-2 mutations and uneven vaccination coverage among populations, a series of COVID-19 waves have been caused by different variants of concern (VOCs). Despite the updated vaccine formulations for the new VOC, the benefits of additional COVID-19 vaccine doses have raised many doubts, even among high-risk groups such as healthcare workers (HCWs). We examined the factors underlying hesitancy to receive COVID-19 booster vaccine doses and analysed the anti-SARS-CoV-2 IgG antibody response after booster vaccination among HCWs. Our study found that 42% of the HCWs were hesitant about the second booster dose, while 7% reported no intent to get vaccinated with any additional doses. As reasons for not vaccinating, participants most frequently highlighted lack of time, negative experiences with previous vaccinations, and immunity conferred by past infections. In addition, we found the lowest post-vaccination antibody titres among HCWs who did not receive any vaccine booster dose and the highest among HCWs vaccinated with two booster doses.

## 1. Introduction

According to the Hippocratic oath ‘*Morbum evitare quam curare facilius est*’, it is better to prevent than to treat disease. The best way to prevent the spread and pathogenesis of infectious disease agents is vaccination. There are different types of vaccines, which may contain attenuated, inactivated or dead pathogens, as well as purified products derived from them or fragments of their genetic material encoding the antigens [1]. In addition, there are different vaccination schemes that differ in their number of doses, the time between each dose, the administration route, etc. [1]. Lastly, the medical recommendations regarding vaccinations may change due to newly acquired knowledge [2].

According to the Centers for Disease Control and Prevention (CDC), four million deaths worldwide are prevented by childhood vaccination every year [3]. Moreover, it was estimated that more than 50 million deaths could be prevented through immunisation between 2021 and 2030 [3]. Over the next 10 years, measles and hepatitis B vaccinations can save nearly 19 and 14 million lives, respectively [3]. Finally, it was estimated that vaccinations prevented almost 20 million deaths from coronavirus disease 2019 (COVID-19) worldwide in the first year after vaccine approval [4].

However, tackling the vaccine-preventable diseases is mostly limited by vaccine hesitancy. In January 2019, even before the COVID-19 pandemic outbreak, the World Health Organisation (WHO) identified vaccine hesitancy among ten threats to global health [5]. Importantly, they also pointed out three groups of viruses that may pose pandemic or epidemic risk such as influenza viruses, Ebola viruses (and other high-threat pathogens like several haemorrhagic fevers, Zika, Nipah, Middle East respiratory syndrome coronavirus (MERS-CoV), and Severe Acute Respiratory Syndrome coronavirus (SARS-CoV), as well as unknown emerging infectious disease pathogens), and human immunodeficiency virus (HIV) [5]. Although vaccination plays a crucial role in preventing viral infections and transmission, the reasons for vaccine hesitancy remain unclear. WHO research recognised complacency, vaccine access inconvenience and general distrust as main factors influencing vaccine acceptance [5]. As a result, there is a huge discrepancy between the vaccination coverage recommended by WHO and the actual rate of vaccination. For example, in 2003 the WHO’s plan for influenza vaccination was to achieve 50% vaccination coverage among the elderly by 2006 and increase it to 75% by 2010 [5]. In 2009, the European Council recommended increasing the vaccination rate among risk groups, including healthcare workers (HCWs) [6]. Research by Sprujit et al. on 14 European countries until the 2013/2014 season revealed that median influenza vaccination rates in the general population were low overall, whereas rates among the elderly were higher and close to the threshold recommended by the WHO(1% to 27% and 2% to 81%, respectively). Regardless of the study period, ranging from 8 to 23 seasons in different countries, the observed vaccination coverage trend rose slightly initially, but began to decline after the 2009 A/H1N1 pandemic year. This pattern was observed in both the total population and the elderly [7]. A global meta-analysis of influenza vaccination showed the highest vaccination rates in the general population during the COVID-19 pandemic (27.63%), followed by other influenza seasons (25.48%) and the 2009 A/H1N1 pandemic (20.41%). Similarly, the largest number of vulnerable people was vaccinated during the COVID-19 pandemic (48.52% of HCWs and 54.59% of the chronically ill), but none of them reached the WHO target. The opposite trend was observed for other periods, with more HCWs and chronically ill persons vaccinated during the 2009 A/H1N1 pandemic (38.98% and 42.75%, respectively) than other influenza seasons (33.95% and 40.69%, respectively) [8]. In Poland, based on the National Institute of Public Health—National Hygiene Institute and vaccine distributors’ data, it is estimated that since the 2011/2012 season the influenza vaccination rate was the highest during the COVID-19 pandemic and reached its peak in 2021–2022, with 7% of the general population and 22.9% of the elderly vaccinated, but after this year it started declining. From the 2011/2012 season to the 2018/2019 season the vaccination coverage slowly declined and started increasing in the 2017/2018 season. Nevertheless, a low median vaccination rate was observed during this time in both the general population (3.7%) and the elderly (12.1%), which did not change considerably (by 1.2% in the general population and 2.8% among the elderly) [9].

Overall, the obtained data show discrepancies between vaccination coverage in general and susceptible groups, as well as in different countries [7,8]. However, the lack of detailed official data from each country [5,10] only provides general conclusions, but does not allow us to tailor vaccination campaigns to specific countries and groups. As indicated by WHO, influenza vaccination uptake is related to different factors that are strongly country and context-dependent [5]. This highlights the need to monitor vaccination rate and hesitancy at specific settings, with particular attention to HCWs, in order to address the problem of vaccination coverage.

One year after the emergence of severe acute respiratory syndrome coronavirus 2 (SARS-CoV-2) and its global spread, there were four COVID-19 vaccines available in the European Union: BNT162b2 (Pfizer, New York, NY, USA—BioNTech, Mainz, Germany), mRNA-1273 (Moderna, Cambridge, MA, USA), and ChAdOx1-S (AstraZeneca, Cambridge, UK/University of Oxford, Oxford, UK), which were required to be administered in two doses, and one shot Ad26.COV2.S (Janssen Pharmaceutical Companies, Beerse, Belgium). Due to the anti-SARS-CoV-2 antibody waning [11] and relatively little knowledge on the role of memory cell in the re- or post-vaccination infection, the European Centre for Disease Control and Prevention (ECDC) and European Medical Agency (EMA) in 2022 recommended the first booster dose mainly for the individuals with severe COVID-19 high risk, such as elderly individuals, as well as for people with high risk of SARS-CoV-2 infection, i.e., HCWs [10]. Furthermore, due to the emergence of the SARS-CoV-2 omicron genetic variant with the ability to escape the immune system, a second booster dose was recommended [10]. Currently in Poland, there are three booster vaccines available: BNT162b2 BA.4-5 (Pfizer, New York, NY, USA—BioNTech, Mainz, Germany), mRNA-1273 BA.1 (Moderna, Cambridge, MA, USA) and NVX-CoV2373 (Novavax, Gaithersburg, MA, USA).

In our previous study [12], we assessed anti-SARS-CoV-2 S Ab levels after completion of a mandatory COVID-19 primary vaccination series among HCWs in February 2021. Within seven months of vaccination we observed that the Ab level decreased by approximately 90–95%; however, none of the HCWs contracted COVID-19. During that period, Alpha VOC had been the major lineage in Poland and was replaced by the Delta VOC in July 2021 [13]. In addition, we revealed that early in the pandemic (September 2020) none of the HCWs was infected with SARS-CoV-2, regardless of the occupational risk for SARS-CoV-2 infection. Interestingly, at the end of the pre-VOC time (December 2020), almost a two times-higher seroprevalence was detected among the HCWs from the low infection risk unit than from the high infection risk unit. Thus, we concluded that awareness and training of HWCs from the infectious disease unit may contribute to the reduction of SARS-CoV-2 transmission in healthcare settings, even in the absence of a vaccine. Considering both the rapid waning of vaccination-induced anti-SARS-CoV-2 S Ab levels after the primary vaccination schedule and the effectively implemented non-pharmaceutical interventions, we aimed to re-examine the attitude of the HCWs from the infectious disease unit toward a non-compulsory booster vaccination. The updated booster vaccines, released by Pfizer—BioNTech and Moderna, were designed to target the rapidly spreading Omicron VOC by producing not only S protein from the original strain of SARS-CoV-2, but also from the Omicron BA.1 subvariant or both the Omicron BA.4 and the Omicron BA.5 subvariants of SARS-CoV-2 [10]. Bivalent vaccines were available for HCWs in Poland in September 2021, two months earlier than for the general population [9]. Since February 2022, the Omicron VOC has expanded in Poland, accompanied with surges of new COVID-19 cases at the highest level during the COVID-19 pandemic to date [13]. In response to the increased transmissibility of the Omicron VOC and the higher risk of contracting severe COVID-19 for those in vulnerable groups, the second booster dose in Poland was firstly available for the elderly over 80 years of age in April 2022, followed by HCWs in August 2022, and people over 12 years of age one month later [9]. Here, we present the follow up study which focuses on the anti-SARS-CoV-2 Ab titres after booster immunisation and the COVID-19 vaccine booster hesitancy among HCWs from the high infection risk healthcare unit.

## 2. Materials and Methods

### 2.1. Study Participants and Design

We invited HCWs from the Department of Infectious Diseases and Child Neurology (DIDaCN) and collaborating units, K. Jonscher’s Clinical Hospital, Poznan University of Medical Sciences, Poznan to participate in our study. The invitation was sent by email to all HCWs with basic information on the aim and objectives of this study. In addition, the invitation was spread among HCWs during daily routine meetings. We used only one inclusion criterium, i.e., high risk of SARS-CoV-2 infection connected to the work with COVID-19-positive patients. There were no exclusion criteria.

All participants were asked to answer an online epidemiological survey to collect demographic (sex, age, and profession) and health status (chronic disease and previous SARS-CoV-2 infection) data. Moreover, since receiving COVID-19 booster doses was publicly encouraged, but not mandatory for HCWs in Poland, we asked them whether they received vaccine boosters, how many doses were administered, as well as what reasons underlying vaccination hesitancy. Interviewees could choose from a list of motives for their vaccine hesitancy (i.e., no trust in the vaccine safety; some medical reasons; no time; previous SARS-CoV-2 infection and its severity) and/or fill in their own answers. To avoid additional bias related to the availability of the COVID-19 vaccine booster doses, we invited the study participants to fill the online survey and donate blood at a time when the booster vaccination was fully available and the time from registration to vaccination was shorter than one day.

Blood specimens were drawn by trained nurses at the DIDaCN in November 2022, i.e., approximately one year after the first COVID-19 vaccine booster and two months after the second vaccine booster dose administration. Following collection, the blood samples were transported to the IBCH PAS for serological assays. In order to differentiate between the Ab response induced by natural infection or vaccination we used immunoassays targeting two SARS-CoV-2 proteins: spike (S) and nucleocapsid (NCP). COVID-19 vaccines are based on S protein, so the increased anti-S Ab level is observed in both vaccinated and SARS-CoV-2 infected persons. On the contrary, anti-NCP Ab are produced only after SARS-CoV-2 infection and their elevated level is the indicator of previous natural exposure to SARS-CoV-2

### 2.2. Laboratory Analysis

Serological testing for IgG Ab to the S protein and NCP protein was performed using a quantitative anti-SARS-CoV-2 QuantiVac IgG ELISA test (EuroImmun GmbH, Lübeck, Germany) and anti-SARS-CoV-2 NCP IgG ELISA test (EuroImmun GmbH, Lübeck, Germany), respectively. The presence of anti-S IgG Ab confirmed the response to either prior SARS-CoV-2 infection or vaccination, while the presence of anti-NCP IgG Ab indicated only previous SARS-CoV-2 infection. Assays were carried out and interpreted as described in the manufacturer’s manuals. Results were calibrated into WHO international units (binding antibody unit, BAU/mL) using the attached reference material and calibration curve. For each run, a new calibration curve was obtained and used for calculations. Analysis included HCWs with anti-SARS-CoV-2 IgG Ab measurement within 389±29 days after the first or 44 ± 11 days after their second BNT162b2 booster vaccination. The time between the completing of primary homologous BNT162b2 vaccination series and anti-SARS-CoV-2 IgG Ab measurements was more than 18 months for HWCs who did not receive any BNT162b2 booster dose.

### 2.3. Statistical Analysis

The categorical variables were presented as counts and percentages, and the seroprevalence estimates were presented with the 95% CI calculated using the hybrid Wilson/Brown method. Further analyses were conducted with two-way ANOVA with the Tukey post-hock test with the following within-subject factors: sex (female or male); age (<30, 31–40, 41–50, 51–60 or >60); occupation (physician, nurse/midwife or other); COVID-19 history (positive or negative); chronic diseases (yes or no); and anti-SARS-CoV-2 S IgG level (no vaccine boosters, one dose or two doses of vaccine boosters). The interaction between analysed factors was not included. Data were accepted as statistically different if *p* < 0.05. All statistical analyses were performed using GraphPad Prism 10 software.

### 2.4. Ethics Approval

The study was approved by the Bioethics Committee at the Poznan University of Medical Sciences, Poznan, Poland (Resolution No. 470/20 from 17 June 2019). In addition, written informed consent was obtained from each study participant before blood collection.

## 3. Results

### 3.1. Characteristics of the Study Participants

The study group included 69 HCWs (Table 1), of whom the vast majority took part in our previous study on the anti-SARS-CoV-2 antibody level [12]. Most participants were women (84.06%) and were aged 41.44 ± 13.71 years. Nurses and midwives represented 43.48% of study participants, followed by physicians (27.53%), other health associate professionals (20.29%), and administration staff (8.7%) (Table 1). At the time of enrolment, 47 participants (68.12%) reported a previous laboratory-confirmed SARS-CoV-2 infection and 31 (44.93%) suffered from at least one comorbidity (Table 1).

### 3.2. Prevalence of Anti-SARS-CoV-2 Antibodies among Healthcare Workers

For this project, we analysed the prevalence of anti-SARS-CoV-2 antibodies after natural infection (anti-NCP) and after the anti-COVID-19 BNT126b2 vaccination (anti-S). A positive result of a SARS-CoV-2 diagnostic test in the past was reported by 47 study participants. Among them, we found anti-SARS-CoV-2 NCP antibodies in 20 individuals. In 27 HCWs, the anti-SARS-CoV-2 NCP antibodies were depleted. Seven individuals had developed anti-SARS-CoV-2 NCP antibodies without declaring COVID-19 in their history. The prevalence of anti-SARS-CoV-2 NCP antibodies increased from 5.71% in September 2021 [12] to 39.1% in November 2022.

Post-vaccination antibodies were found in 68 of the 69 analysed HCWs, and the prevalence of anti-SARS-CoV-2 S antibodies increased from 84.50% in September 2021 [12] to 98.56% in November 2022. Among these HCWs, there was one individual with a complete depletion of anti-SARS-CoV-2 S antibodies. The lowest anti-SARS-CoV-2 S antibody levels, i.e., 836.60 ± 686.04 BAU/mL, were found among individuals who had not received any vaccine booster dose (Figure 1). The highest post-vaccination antibody titre, i.e., 3990.17 ± 2053.16 BAU/mL, was observed in study participants who received two COVID-19 vaccine boosters (Figure 1).

### 3.3. The Vaccine Boosters among Healthcare Workers

At the time of enrolment, all HCWs had received the recommended two basic doses of BNT162b2 vaccines (Pfizer—BioNTech) in January/February 2021. The vast majority of study participants (92.75%) had received at least one dose of vaccine booster, with two doses of vaccine booster administered to 50.73% of the analysed individuals (Table 1). Interestingly, all HCWs with no previously confirmed SARS-CoV-2 infection had received at least one vaccine booster dose. Only 5 of 69 HCWs had not received any vaccine booster (Table 1). The highest number of HCWs without any vaccine boosters was observed among administration personnel (Table 1).

Participants in the 31–40 and 41–50 age groups were the most sceptical about the second booster dose (75% and 62%, respectively). Booster hesitancy among health professionals (physicians, nurses and midwives) was lower than among administrative staff and others. Almost 79% of the physicians had received two COVID-19 vaccine booster doses. However, apart from physicians, about half of the HCWs from each occupation group were hesitant about the second booster dose.

In total, 34 of the 69 HCWs provided reasons for COVID-19 booster vaccination hesitancy (Figure 2). Interviewees reported more than one reason. Three of five participants who did not receive any booster shots and 5 of 29 who had received only one booster preferred not to comment on their unwillingness to get the booster doses. Four HCWs planned to get the second booster dose once they were eligible for vaccination.

Concerns expressed by two participants who did not get any booster dose were based on their personal perceptions of the COVID-19 vaccine and disease prevention. They reported negative experiences with past COVID-19 vaccination and stated that the natural immunity developed after SARS-CoV-2 infection could protect them against COVID-19, which, overall, does not pose serious health risks.

Responses from HCWs who received only one COVID-19 booster dose can be categorised into two themes: (i) influences arising from personal perceptions of the COVID-19 vaccine and disease prevention and (ii) issues directly related to vaccination and its safety. Six individuals reported negative adverse effects after past COVID-19 vaccinations. Some believed in the protective effect of previous exposure to SARS-CoV-2, either as a result of infection or direct contact with a SARS-CoV-2-infected person. However, the majority of participants stated that they did not have time to comply with the recommended vaccination schedule. Four HCWs expressed concerns about safety issues, three interviewees mentioned other medical reasons, and one person distrusted vaccine safety in general.

## 4. Discussion

Studies on waning humoral immunity after COVID-19 vaccination showed a reduction in anti-SARS-CoV-2 S Ab levels within seven months of completion of the COVID-19 primary vaccination course, ranging from 72% to 95%. Importantly, Ab levels were higher in previously infected HCWs [12,14]. The first booster dose was demonstrated to significantly increase Ab levels (even up to almost 110%) [15], as well as bolster protection against SARS-CoV-2 infection [16,17,18,19,20]. However, the waning effectiveness of the first booster dose within a few months of administration was reported by Patalon et al. [16]. The second booster dose provided a further increase in anti-SARS-CoV-2 S Ab levels. It is noteworthy that peak responses after the second booster dose were similar to, and possibly better than, peak responses after the first booster dose [21]. In addition, immunity waned more quickly after the second booster than after the first booster dose [21]. Several studies from Israel have demonstrated that the second booster dose increases protection against SARS-CoV-2 infection compared to the first booster dose [19,22]. However, similar protective effects for both the first and second booster doses were observed in patients with cancer in Singapore [23]. Our study showed that the anti-SARS-CoV-2 S Ab in HCWs vaccinated with one booster dose were still detectable and no new SARS-CoV-2 infection occurred ten months after booster vaccination. However, the Ab level was significantly lower than among HCWs with two booster shots. The question that should be answered, but will remain open, is what level of anti-SARS-CoV-2 is effective against severe COVID-19? Only further research may confirm the importance of booster vaccination in the fight against emerging coronaviruses.

One of the main goals of this study was to determine the COVID-19 vaccine booster acceptance among HCWs in Poland. Receipt of at least one booster dose was reported by 92.8% HCWs in October 2022, with more than two in five (42%) HCWs hesitant to receive the second booster dose. Research on the representative polish population conducted at the same time by Sobierajski et al. revealed that half of the respondents (50.6%) received at least one booster dose [24]. Although it differs from the ECDC data from October 2022, according to which the uptake of the first booster in Poland was 39.2%, [25], the evidence we found points to around two times higher acceptance of COVID-19 vaccine booster doses among HCWs than non-HCWs. The observed trend was also reported globally in June 2022 by Lazarus et al. (19.9% vs. 40.3%, respectively) [26].

Likewise, studies carried out in the USA in 2022 demonstrated a higher uptake of the COVID-19 vaccine booster among HCWs than non-HCWs, but the difference between them was smaller. The acceptance rate reported by Agaku et al. in January 2022 among adults working in hospitals was 60.5%, compared to 48.5% among the general population [27]. In March 2022, Lu et al. found that COVID-19 booster dose coverage was 67.4% among essential healthcare personnel, which was comparable to 63.4% among the general population [28]. Similarly, as reported by Farah et al. in April 2022, more than two-thirds of HCWs (64.8%) received a COVID-19 booster vaccination [29]. High levels of acceptance of at least one booster dose among HCWs were also observed in November 2021 in Czechia (71.3%) [30] and in Singapore in December (73.8%) [31]. Strikingly different from previous results reported in the literature are data from Albania, where in June 2022 only 19.1% of HCWs had received a booster dose [32]. These worldwide differences in acceptance of COVID-19 booster doses among HCWs may be a result of country-specific COVID-19 booster vaccine availability and policies, as well as the current epidemiological situation. However, as HCWs are prioritised for COVID-19 booster vaccination in many countries, understanding the problem of vaccine hesitancy is crucial to combat the COVID-19 pandemic and was addressed in our study.

We found that even in the same healthcare unit, COVID-19 booster vaccination hesitancy differed among occupation groups of HCWs. The lower hesitancy among physicians, nurses, and midwives than for non-health professionals reported in our research is consistent with that of other studies [29,32]. Furthermore, the highest COVID-19 vaccine booster uptake among physicians compared to other HCWs also supports previous findings [29,32]. Tendencies in occupational groups of HCWs toward hesitancy to the first dose booster dose continued for the second booster dose. Moreover, the decreasing level of second booster uptake observed in our study is in line with the analysis of vaccination trends in Poland until January 2023 by Walkowiak et al. [33]. At the beginning of the Omicron VOC wave, the huge majority of HCWs showed a dramatic change in their attitude toward vaccination, including the resignation of the second booster dose uptake, whereas the decision-making of accepting was influenced by social impact, infection trends, as well as the availability of updated booster doses [33]. In addition, our results are consistent with those of Galanis et al., who found that around a third of the nurses in Greece were hesitant about the second booster dose. Increased vaccine hesitancy was associated with uncertainty about updated boosters and COVID-19 vaccination in general, and interestingly, with greater compliance with hygiene measures [34]. The latter reason, which was not mentioned by nurses in our present study, concurs well with our previous finding. In the pre-COVID-19 vaccination era HCWs from the surveyed unit effectively complied with non-pharmaceutical interventions, which resulted in a lower seroprevalence than in low-infection risk unit [12]. Since some HCWs preferred not to admit what influenced their vaccine hesitancy, it can be assumed that some of them fell into complacency, which may be supported by their surviving a SARS-CoV-2 infection. To sum up, it seems that the working environment has an uneven impact on COVID-19 vaccine behaviour among HCWs from different occupational groups, and other factors influencing vaccine hesitancy need to be taken into account. With respect to other personal motives underlying vaccine hesitancy, we found that previous SARS-CoV-2 infection and negative experiences following COVID-19 vaccination were reasons for hesitancy among those who had not received the second booster dose or either booster dose. Hesitancy among HCWs due to the fear of side effects after booster vaccination was also reported by Dziedzic et al. [35]. In addition, hesitancy about any booster dose arose from the belief that COVID-19 is low risk, which is surprising given that the participants worked in an infectious disease unit. Reluctance to receive the second booster dose was mostly associated with a lack of time.

Despite the proven safety of the mRNA vaccines given as a second booster dose [15,21], safety concerns also prevented HCWs from getting vaccinated. Misinformation not only about the COVID-19 vaccine but also about other vaccines in general undermine vaccine confidence and should be addressed by the scientific community [36,37,38,39].

Overall, our results corroborate the findings on attitudes towards COVID-19 booster vaccination among adult Poles conducted by Rzymski et al. in September 2021. They established that the main reasons for vaccine hesitancy were side effects experienced after previous doses, the opinion that further vaccination was unnecessary, and safety issues [40]. It should be noted that people often do not fully express their fears regarding COVID-19 booster vaccination and in our study, we also found HCWs who did not want to comment on their unwillingness to receive a booster dose. Importantly, as highlighted by Sobierajski et al., public declarations may not align with vaccine behaviour, especially among moderate vaccine supporters/opponents, and should be interpreted with caution [24]. For example, the declared willingness to receive a COVID-19 booster dose in Poland was 71% in September 2021 [40] and 84.2% in July 2022 [26], while according to ECDC data, the uptake of the first booster in Poland in July 2023 was 39.3% [25].

Notwithstanding the relatively small sample, this work offers valuable insights into COVID-19 booster uptake and hesitancy among HCWs in Poland. As being prioritised for booster vaccination does not appear to be a sufficient incentive, close attention should be paid to misinformation in the media, which may shape the COVID-19 vaccine beliefs and behaviour of HCWs as well as the general population. Further research on anti-SARS-CoV-2 S Ab levels following additional booster dose vaccinations would be of great help towards revising vaccination policies to control the outbreak of future waves of COVID-19.

## 5. Conclusions

Vaccination is the best method to prevent infectious diseases. However, due to many factors, such as anti-vaccination activism, fake news, and the lack of a clear and easy to follow vaccination campaign, many in society are reluctant to receive booster doses of COVID-19 vaccines. Healthcare workers are at the highest risk of SARS-CoV-2 infection, and thus should be well-educated and understand the importance of vaccine booster doses for the control and prevention of COVID-19. We found that almost all HCWs from the high SARS-CoV-2 infection risk unit were vaccinated with a first BNT162b2 booster dose, but more than two in five were hesitant to receive the second booster dose. The presence of the anti-SARS-CoV-2 antibodies, previous negative experiences with vaccination, beliefs about the protective effect of prior infections, and a lack of time were the most common barriers to booster uptake reported by HCWs. Our study shows that information about the effectiveness and safety of the updated COVID-19 vaccines should be widely campaigned among HCW, who may not be keeping updated regarding new knowledge and evidence about benefits of the vaccines in the face of rapidly evolving VOC. We believe that our findings will serve as a basis for policy-makers to increase the booster vaccination coverage not only among HCWs, but also the general population.

## Figures and Tables

**Figure 1 vaccines-12-00475-f001:**
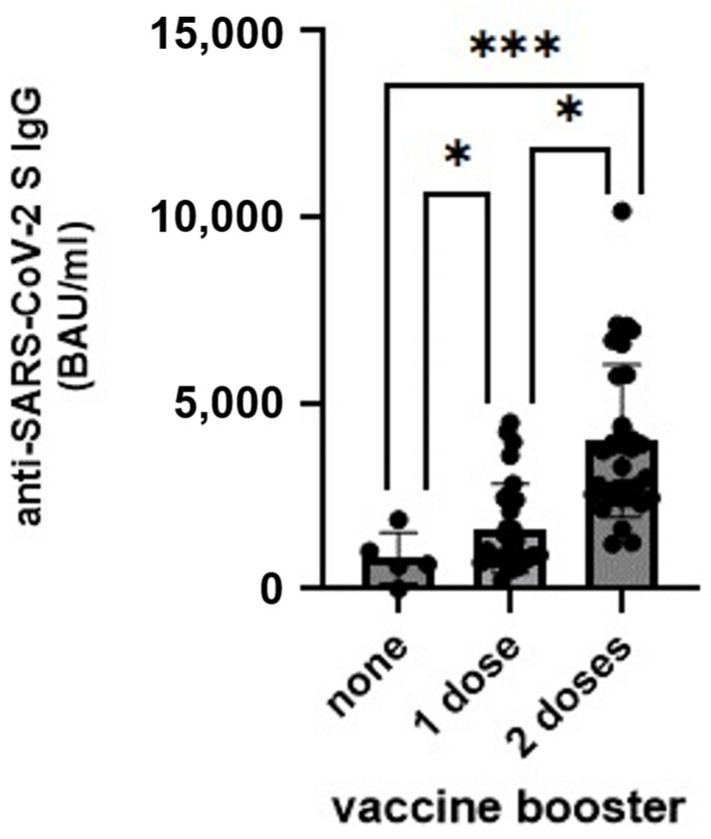
The anti-SARS-CoV-2 S antibodies level among healthcare workers who received none, one or two doses of COVID-19 vaccine boosters. Statistical analyses were conducted with two-way ANOVA with the Tukey post-hock test. BAU—binding antibody units, each dot represents one study participant, *—*p* < 0.05, ***—*p* < 0.001.

**Figure 2 vaccines-12-00475-f002:**
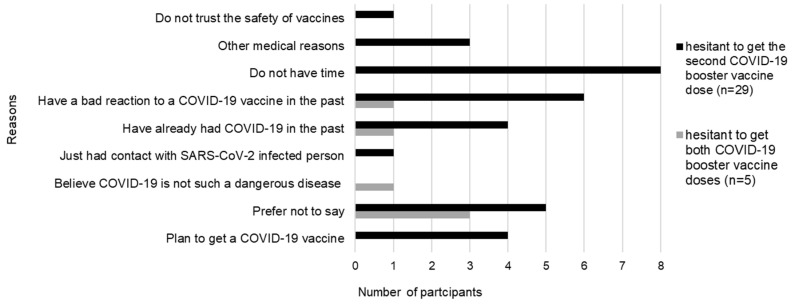
The reasons behind the hesitancy to receive the second (n = 29) or both (n = 5) COVID-19 booster vaccine doses among healthcare workers. Participants could report more than one reason.

**Table 1 vaccines-12-00475-t001:** The COVID-19 booster vaccination among healthcare workers with the highest SARS-CoV-2 infection risk.

Category	Participants		Booster Vaccination		
	(n)	(%)	None (%, 95% CI)	One Dose (%, 95% CI)	Two Doses (%, 95% CI)
**Overall**	69	100	7.25 ^a^ (2.39–16.11)	42.02 ^b^ (30.24–54.52)	50.73 ^b^ (38.41–62.98)
**Gender**					
Female	58	84.06	6.90 ^a^ (1.91–16.73)	50.00 ^b^ (36.58–63.42)	43.10 ^b^ (30.16–56.77)
Male	11	15.94	9.09 ^a^ (0.23–41.28)	0.00 ^ab^ (0.00–28.49) *	90.91 ^b^ (58.72–99.77)
**Age**					
19–30 y.o.	20	28.99	10.00 ^a^ (1.24–31.07)	35.00 ^ab^ (15.39–59.22)	55.00 ^b^ (31.53–76.94)
31–40 y.o.	12	17.39	8.33 ^a^ (0.21–38.48)	16.67 ^ab^ (2.09–48.41)	75.00 ^b^ (42.81–94.51)
41–50 y.o.	13	18.84	7.69 (0.19–36.03)	61.54 (31.58–86.14)	30.77 (9.09–61.43)
51–60 y.o.	20	26.09	5.00 ^a^ (0.13–24.87)	25.00 ^ab^ (8.66–49.10)	70.00 ^b^ (45.72–88.11)
>60 y.o.	4	8.70	0.00 (0.00–60.24)	0.00 (0.00–60.24)	100.00 (39.76–100.00) *
**Occupation**					
Physicians	19	27.53	5.26 ^a^ (0.13–26.03)	15.79 ^ab^ (3.38–39.58)	78.95 ^b^ (54.43–93.95)
Nurses/Midwives	30	43.48	6.67 ^a^ (0.82–22.07)	53.33 ^b^ (34.33–71.66)	40.00 ^b^ (22.66–59.40)
Administration	6	8.70	16.67 (0.42–64.12)	50.00 (11.81–88.19)	33.33 (4.33–77.72)
Others	14	20.29	7.14 (0.18–33.87)	42.86 (17.66–71.14)	50.00 (23.04–76.96)
**COVID-19 history**					
Positive results	47	68.12	10.64 ^a^ (3.55–23.10)	38.30 ^b^ (24.51–53.62)	51.06 ^b^ (36.06–65.92)
Negative results	22	31.88	0.00 ^a^ (0.00–15.44) *	50.00 ^b^ (28.22–71.78)	50.00 ^b^ (28.22–71.78)
**Chronic diseases**					
Yes	31	44.93	6.45 ^a^ (0.79–21.42)	38.71 ^b^ (21.85–57.81)	54.84 ^b^ (36.03–72.68)
No	38	55.07	7.89 ^a^ (1.66–21.38)	44.74 ^b^ (28.62–61.70)	47.37 ^b^ (30.98–64.18)

* one-sided 97.5% confidence interval; ^a,b^—values within a row with different superscript letters differ *p* < 0.05.

## Data Availability

The data presented in this study are available on request from the corresponding author. The data are not publicly available due to privacy restrictions.

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
