# Peer review of "Anti-SARS-CoV-2 Antibodies Level and COVID-19 Vaccine Boosters among Healthcare Workers with the Highest SARS-CoV-2 Infection Risk—Follow Up Study"

_vaccines, 2024, doi:10.3390/vaccines12050475_

Round 1
Reviewer 1 Report (Previous Reviewer 1)
Comments and Suggestions for Authors
This well designed, conducted and discussed study examined factors underlying hesitancy to receive COVID-19 booster vaccine doses and analyzed the anti-SARS-CoV-2 IgG antibody response after booster vaccination. They study reports that 42% of the HCWs were hesitant about the second booster dose, while 7% reported no intent to get vaccinated with any additional doses. Reasons for not vaccinating are well surveyed and discussed. The revised paper addressed minor queries that improved readability and inclusion for systematic reviews studies.
Author Response
We are grateful for your positive feedback!
Reviewer 2 Report (Previous Reviewer 3)
Comments and Suggestions for Authors
This manuscript extends data from the previous work (ref no. 12) that describes the antibody levels and hesitancy among healthcare workers. The data could be extending to make it more informative in the Figure 1.
Comments.
1. Line 48: Suggest using "SARS-CoV-1" to make it consistent with MERS-CoV.
2. Line 93: Suggest using "ChAdOx1-S" instead of "ChAdOx1" because ChAdOx1 is only a viral vector platform, not an embedded spike gene of SARS-CoV-2.
3. Figure 1: Suggest clarifying the type of central tendency with error (line and error bars) and statistical test to the figure description. Moreover, the "Results" did not show this value from Figure 1 in the body text. I suggest adding more information about the IgG levels in section 3.2 to make it more informative.
4. Figure 1: You can subgroup by the COVID-19 history and compare it to make your data more insightful.
Typos.
1. Line 104: "BA.4-5". (must include the dot)
Author Response
Q1 – This manuscript extends data from the previous work (ref no. 12) that describes the antibody levels and hesitancy among healthcare workers. The data could be extending to make it more informative in the Figure 1.
Comments. 1. Line 48: Suggest using "SARS-CoV-1" to make it consistent with MERS-CoV.
A1 – We would like to thank the referee once more for sparing the time to write so many useful and detailed comments. We modified the virus name abbreviation.
Q2 – 2. Line 93: Suggest using "ChAdOx1-S" instead of "ChAdOx1" because ChAdOx1 is only a viral vector platform, not an embedded spike gene of SARS-CoV-2.
A2 – Thank you for your comment, we corrected the name.
Q3 – 3. Figure 1: Suggest clarifying the type of central tendency with error (line and error bars) and statistical test to the figure description. Moreover, the "Results" did not show this value from Figure 1 in the body text. I suggest adding more information about the IgG levels in section 3.2 to make it more informative.
A3 – Thank you for your suggestion. The Figure 1 contains scatter plot with error bars, performed in the GraphPad Prism software. WE have added some information on the statistical test to the figure legend. The antibodies levels were added into the manuscript.
Q4 – 4. Figure 1: You can subgroup by the COVID-19 history and compare it to make your data more insightful.
A4 – Thank you for your comment. We have already analysed the anti-SARS-CoV-2 S antibodies in study participants with different COVID-19 history and showed that the individuals with severe COVID-19 have the highest antibodies level in our previous manuscript (Lorent et al., Vaccines, 2022). Such analysis within this follow up study would be unnecessary repetition in our opinion.
Q5 – Typos. 1. Line 104: "BA.4-5". (must include the dot)
A5 – Thank you for your comment, we corrected the spelling.
Reviewer 3 Report (Previous Reviewer 4)
Comments and Suggestions for Authors
Whilst I would not totally agree with the comment of the author stating that the paper is innovative, there is no doubt that the manuscript is stronger for the amendments made to the text. Therefore, I would recommend publication
Author Response
Thank you for your positive feedback!
This manuscript is a resubmission of an earlier submission. The following is a list of the peer review reports and author responses from that submission.
Round 1
Reviewer 1 Report
Comments and Suggestions for Authors
This follow-up study provides interesting insights into hesitancy toward uptake of COVID-19 booster among healthcare workers. They followed 69 healthcare workers who received first dose of vaccine on their uptake of first and second vaccine booster. Majority (~51%) received two doses of booster vaccine while 7.5% have opted not getting any protection from booster vaccination. Given significantly higher level of anti-SARS-CoV-2 S antibodies detected in sera from subjects vaccinated twice with booster; authors pointed to surprisingly high percentage (42%) of participants who hesitated to take second dose of booster. From the survey they listed several reasons for this hesitancy toward boosting effect of vaccination namely side effects experienced after previous doses, the opinion that further vaccination was unnecessary, and safety issues. This is well written paper that completes earlier study from the same group and expands observations toward booster vaccination in the time of decreased concerns/fears of COVID19 infection. There are several small queries that if addressed may improve this study:
1. Given relatively small sample (69 participants) and high number of women (58) it would be interesting to provide gender data for 19 physicians in this study of which 78% received second booster.
2. Provide comment on decreasing level of second booster uptake based for different groups of healthcare workers (Physicians-78%, Nurses-40%, Administration-33%).
3. Consider removing paragraph in discussion on linking hesitancy toward booster vaccination with shortage of doctors in polish healthcare market and perceived lack of time.
Comments on the Quality of English LanguageReplace “any” to “none" in legend in tables and figures for participants who have not received any booster vaccine.
Author Response
Q1 – This follow-up study provides interesting insights into hesitancy toward uptake of COVID-19 booster among healthcare workers. They followed 69 healthcare workers who received first dose of vaccine on their uptake of first and second vaccine booster. Majority (~51%) received two doses of booster vaccine while 7.5% have opted not getting any protection from booster vaccination. Given significantly higher level of anti-SARS-CoV-2 S antibodies detected in sera from subjects vaccinated twice with booster; authors pointed to surprisingly high percentage (42%) of participants who hesitated to take second dose of booster. From the survey they listed several reasons for this hesitancy toward boosting effect of vaccination namely side effects experienced after previous doses, the opinion that further vaccination was unnecessary, and safety issues. This is well written paper that completes earlier study from the same group and expands observations toward booster vaccination in the time of decreased concerns/fears of COVID19 infection. There are several small queries that if addressed may improve this study:
Given relatively small sample (69 participants) and high number of women (58) it would be interesting to provide gender data for 19 physicians in this study of which 78% received second booster.
A1 – Thank you for your comment! Indeed, in our study participants group we have overrepresentation of women. Within mentioned 19 physicians, all male received second booster, while 1 female did not get any booster vaccination, and 3 women were vaccinated only with one booster dose. We are aware that this may lead to some bias, but we would like to highlight that current trend in HCW shows that there are more female than male physicians, and thus we believe that our results may represent the HCWs working in the Infection Units.
Q2 – Provide comment on decreasing level of second booster uptake based for different groups of healthcare workers (Physicians-78%, Nurses-40%, Administration-33%).
A2 – Thank you for your comment. We modified the Discussion section and the comment according to your suggestion.
‘Tendencies in occupational groups of HCWs toward hesitancy to the first dose booster dose continued for the second booster dose. Moreover, the decreasing level of second booster uptake observed in our study is in line with the analysis of vaccination trend in Poland until January 2023 by Walkowiak et al [27]. At the beginning of Omicron VOC wave, the huge majority of HCWs showed a dramatic change in their attitude toward vac-cination, including resignation of the second booster dose uptake, whereas deci-sion-making of accepting was influenced by social impact, infection trends, as well as the availability of updated booster doses [27]. In addition, our results are consistent with those of Galanis et al., who found that around a third of the nurses in Greece were hesitant about the second booster dose. Increased vaccine hesitancy was associated with uncer-tainty about updated boosters and COVID-19 vaccination in general, and interestingly, with greater compliance with hygiene measures [28]. The latter reason, which was not mentioned by nurses in our present study, concurs well with our previous finding. In the pre-COVID-19 vaccination era HCWs from the surveyed unit effectively complied non-pharmaceutical interventions, which resulted in lower seroprevalence than in low infection risk unit [7]. Since some HCWs preferred not to admit what influenced their vac-cine hesitancy, it can be assumed that some of them fell into complacency, which may be supported by surviving a SARS-CoV-2 infection. To sum up, it seems that the working en-vironment has an uneven impact on COVID-19 vaccine behaviour among HCWs from different occupational groups, and other factors influencing vaccine hesitancy need to be taken into account.’
Q3 – Consider removing paragraph in discussion on linking hesitancy toward booster vaccination with shortage of doctors in polish healthcare market and perceived lack of time.
A3 – Thank you for your comments. We deleted the paragraph, as suggested.
Q4 – Comments on the Quality of English Language
Replace “any” to “none" in legend in tables and figures for participants who have not received any booster vaccine
A4 – Thank you for your comment, we corrected the sentences in the legend in the Figure 1, in the Figure 1 and in the heading in the Table 1.
Reviewer 2 Report
Comments and Suggestions for Authors
Thank you for asking me to review this article.
The topic under study is very important, especially in the post-pandemic context in which although the role of vaccination as a powerful life-saving weapon in the fight against infectious diseases has been further highlighted following the introduction of the 2019 coronavirus vaccine, there is still a discrepancy between the scientific evidence on the efficacy of vaccines and the perceived risk attributed to them.
In this regard, investigating the influence of the perception of a health risk on the adoption of the correct preventive measures is a topic of relevant importance for both health surveillance and the implementation of the best prevention strategies both in Poland and the rest of the world.
The analysis of the determinants of vaccination hesitancy in the population with particular reference to healthcare workers represents a fundamental tool for implementing strategic measures aimed at improving information on this issue and increasing adherence to national and international immunisation programmes.
Overall, the methodology of the study is well described, however, in spite of the topicality of the subject under consideration, a Major Revision is deemed necessary before proceeding with a further peer-review process. The following are my suggestions.
In the introduction, the hypotheses that the research intended to test were described through a rich and accurate literature review; however, the topic of vaccine hesitancy is only hinted at in the final section describing the purpose of the work. In my opinion, this represents a gap in the background; indeed, such a sensitive topic, which the World Health Organisation itself describes as one of the top 10 threats to global public health, deserves in-depth study. I would suggest that the phenomenon of Vaccine Hesitancy should be addressed first, and then declined by health workers and in the local context also in terms of vaccination coverage recorded in the pre-pandemic, pandemic and post-pandemic periods. In fact, the observation of a phenomenon such as low vaccine coverage together with the contingent situations linked to context-specific healthcare, can support the analysis of the literature on the topics of 1) adherence to the anti-SARS-CoV-2 vaccine among healthcare workers, 2) the socio-demographic factors associated with the phenomenon of Vaccine Hesitancy in the same cohort and 3) the analysis of the factors that act as possible barriers or facilitators of vaccine compliance.
As far as methodology is concerned, since the one proposed is a study describing secondary analyses, there are references to the previous study within the text. This choice is understandable but, in my opinion, cross-references to other previously published papers risks negatively affecting originality. Perhaps the authors could think about delving into the macro-sections describing the questionnaire administered by the authors in order to offer a general overview to get a clearer idea of the method and devote a paragraph to the study setting where they describe the context in which the survey was carried out. I also suggest spending a few more words on the recruitment method.
Discussion: The results are discussed exploring numerous interpretations. They are described appropriately.
Minor revisions. I suggest that authors insert tables and figures near the text citing them.
Author Response
Q1 – Thank you for asking me to review this article. The topic under study is very important, especially in the post-pandemic context in which although the role of vaccination as a powerful life-saving weapon in the fight against infectious diseases has been further highlighted following the introduction of the 2019 coronavirus vaccine, there is still a discrepancy between the scientific evidence on the efficacy of vaccines and the perceived risk attributed to them. In this regard, investigating the influence of the perception of a health risk on the adoption of the correct preventive measures is a topic of relevant importance for both health surveillance and the implementation of the best prevention strategies both in Poland and the rest of the world. The analysis of the determinants of vaccination hesitancy in the population with particular reference to healthcare workers represents a fundamental tool for implementing strategic measures aimed at improving information on this issue and increasing adherence to national and international immunisation programmes. Overall, the methodology of the study is well described, however, in spite of the topicality of the subject under consideration, a Major Revision is deemed necessary before proceeding with a further peer-review process. The following are my suggestions. In the introduction, the hypotheses that the research intended to test were described through a rich and accurate literature review; however, the topic of vaccine hesitancy is only hinted at in the final section describing the purpose of the work. In my opinion, this represents a gap in the background; indeed, such a sensitive topic, which the World Health Organisation itself describes as one of the top 10 threats to global public health, deserves in-depth study. I would suggest that the phenomenon of Vaccine Hesitancy should be addressed first, and then declined by health workers and in the local context also in terms of vaccination coverage recorded in the pre-pandemic, pandemic and post-pandemic periods. In fact, the observation of a phenomenon such as low vaccine coverage together with the contingent situations linked to context-specific healthcare, can support the analysis of the literature on the topics of 1) adherence to the anti-SARS-CoV-2 vaccine among healthcare workers, 2) the socio-demographic factors associated with the phenomenon of Vaccine Hesitancy in the same cohort and 3) the analysis of the factors that act as possible barriers or facilitators of vaccine compliance.
A1 – Thank you for your comment. We modified the Introduction section and added comprehensive background to the research according to your suggestion.
‘However, tackling the vaccine-preventable diseases is mostly limited by vaccine hes-itancy. On January 2019, even before the COVID-19 pandemic outbreak, the World Health Organisation (WHO) identified vaccine hesitancy among ten threats to global health. [5] Importantly, they also pointed three groups of viruses that may pose pandemic or epi-demic risk such as Influenza, Ebola (and other high-threat pathogens like several haem-orrhagic fevers, Zika, Nipah, Middle East respiratory syndrome coronavirus (MERS-CoV), Severe Acute Respiratory Syndrome (SARS) as well as unknown emerging infectious dis-ease pathogens), and Human Immunodeficiency Virus (HIV) [5]. Although vaccination plays a crucial role in preventing viral infections and transmission, the reasons for vac-cine hesitancy remains unclear. WHO research recognized complacency, vaccine access inconvenience and general distrust as main factors influencing vaccine acceptance. [5]. As a result, there is a huge discrepancy in vaccination coverage recommended by WHO and obtained rate of vaccination. For example, in 2003 the WHO's plan for influenza vaccina-tion was to achieve 50% vaccination coverage among the elderly by 2006 and increase it to 75% by 2010 [5]. In 2009, increasing vaccination rate among risk groups, including healthcare workers, was recommended by the European Council [6]. A research by Sprujit et al. on 14 European countries until the 2013/2014 season revealed that median influenza vaccination rates in the general population were overall low, whereas rates among the el-derly were higher, even in some countries around the WHO target (1% to 27% and 2% to 81%, respectively). Regardless of the study period, ranging from 8 to 23 seasons in differ-ent countries, the observed vaccination coverage trend was initially slightly rising, but began to decline after the 2009 A/H1N1 pandemic year. This pattern was observed in both total population and the elderly [7]. A global meta-analysis of influenza vaccination showed the highest vaccination rates in general population during the COVID-19 pan-demic (27,63%), followed by other influenza seasons (25,48%) and the 2009 A/H1N1 pan-demic (20,41%). Similarly, the largest number of vulnerable people was vaccinated during the COVID-19 pandemic (48,52% of healthcare workers and 54,59% of chronically ill), but none of them reached the WHO target. The opposite trend was observed for other periods, with more healthcare workers and chronically ill persons were vaccinated during the 2009 A/H1N1 pandemic (38,98% and 42,75%, respectively) than others influenza seasons (33,95% and 40,69%, respectively) [8]. In Poland, based on the National Institute of Public Health – National Hygiene Institute and vaccine distributors’ data, it is estimated that since 2011/2012 season the influenza vaccination rate was the highest during the COVID-19 pandemic and reached the peak in 2021-2022, with 7% of general population and 22,9% of the elderly vaccinated, but after this year started declining. From 2011/2012 season to 2018/2019 seasons the vaccination coverage was slowly declining and started increasing in 2017/2018 season. Nevertheless, a low median vaccination rate was ob-served during this time in both general population (3,7%) and the elderly (12,1%), which did not change considerably (by 1,2% in the general population and 2,8% among the el-derly) [9].
Overall, the obtained data show discrepancies between vaccination coverage in gen-eral and susceptible groups, as well as in different countries [7,8]. However, the lack of detailed official data from each country [5,10] only provides general conclusions, but does not allow to tailor vaccination campaigns to specific countries and groups. As indicated by WHO, influenza vaccination uptake is related to different factors that are strongly country and context-dependent [5]. This highlights the need to monitor vaccination rate and hesitancy at specific settings, with particular attention to healthcare workers, in order to address the problem of vaccination coverage.’
Q2 – As far as methodology is concerned, since the one proposed is a study describing secondary analyses, there are references to the previous study within the text. This choice is understandable but, in my opinion, cross-references to other previously published papers risks negatively affecting originality. Perhaps the authors could think about delving into the macro-sections describing the questionnaire administered by the authors in order to offer a general overview to get a clearer idea of the method and devote a paragraph to the study setting where they describe the context in which the survey was carried out. I also suggest spending a few more words on the recruitment method.
A2 – We are grateful for your valuable comment. As suggested, we deleted the references to the previous study, since it was not relevant in this section. In addition, we added some information on the recruitment method and inclusion/exclusion criteria. The online epidemiological survey was described in details, and the context of the survey was added to the manuscript.
‘We invited HCWs from the Department of Infectious Diseases and Child Neurology (DIDaCN) and collaborating units, K. Jonscher’s Clinical Hospital, Poznan University of Medical Sciences, Poznan to participate in our study. The invitation was sent by email to all HCWs with basic information on the aim and objectives of this study. In addition, the invitation was spread among HCWs during daily routine meetings. We used only one in-clusion criterium, i.e., high risk of SARS-CoV-2 infection connected to the work with COVID-19 positive patients. There were no exclusion criteria.
All participants were asked to answer an online epidemiological survey to collect demographic (sex, age, profession) and health status (chronic diseases, previous SARS-CoV-2 infection) data. Moreover, since COVID-19 booster doses vaccination was publicly encouraged, but not mandatory for HCWs in Poland, we asked them whether they received vaccine boosters, how many doses were administered, as well as what were the reasons underlying vaccination hesitancy. Interviewees could choose from the list of motives for their vaccine hesitancy (i.e., no trust in the vaccine safety; some medical rea-sons; no time; previous SARS-CoV-2 infection and its severity) and/or fill in their own answers.To avoid additional bias related with the availability of the COVID-19 vaccine booster doses, we invited the study participants to fill the online survey and donate blood at the time where booster vaccination was fully available and the time from registration to vac-cination was shorter than one day. ’
Q3 – Discussion: The results are discussed exploring numerous interpretations. They are described appropriately.
A3 – Thank you for your comment.
Q4 – Minor revisions. I suggest that authors insert tables and figures near the text citing them.
A4 – Thank you for your comment. We changed the order accordingly.
Reviewer 3 Report
Comments and Suggestions for Authors
This manuscript extends data from the previous work that delineates the antibody levels and hesitancy among healthcare workers who received the COVID-19 vaccination. This finding is a small-scale study but provides some insight into outcomes. The methodology and outcomes were fine. However, some information needs to be addressed to clarify it.
Major concerns.
1. Lines 82-84: Suggest clarifying the antibody targeting of each test to make it easy to understand to readers. Some readers who are not familiar with the serology will raise the question of why you determined two IgGs.
- Anti-nucleocapsid IgG (NCP)
- Anti-spike IgG (QuantiVac)
Furthermore, the immunologic outcomes, such as Figure 1, were mentioned as "anti-SARS-CoV-2 S IgG", but the Materials and Methods section did not mention what the anti-S is.
2. Suggest adding the "conversion factor" of the QuantiVac assay to the Materials and Methods section because this assessment was not reported initially in the BAU/mL unit.
Comments.
1. Line 46: Suggest using "University of Oxford", a formal form of this university.
2. Line 87: Suggests deleting "(EuroImmun GmbH) "because you were the first mentioned manufacturer of this test (NCP) in line 83.
3. Line 91: Suggest moving "(EuroImmun GmbH)" to line 84 because you were the first mention of this test (QuantiVac) in line 84.
4. Line 88, 119 and elsewhere: Suggest using "BNT162b2" instead of "BNT162b2 mRNA COVID-19" to make it concise. The BNT162b2 is a very specific term that there is no need to mention further to make it lengthy.
5. Lines 126-132: Suggest adding the median or mean interval day between the last dose and the blood draw to ensure the booster dose will increase immunity.
Moreover, non-booster or first booster vaccination may cause the last dose for a long time, giving low immunity levels due to the immunity waning.
Typos.
1. Line 45: "Astra Zeneca". Suggests using "AstraZeneca" (no spacing).
2. Line 57: Suggests checking vaccine name "NVX-CoV2372". I think the FDA-approved vaccine is NVX-CoV2373.
3. Suggest using "Pfizer—BioNTech" instead of "Pfizer/BioNTech" throughout the manuscript. The Pfizer—BioNTech is almost used form.
Author Response
Q1 – This manuscript extends data from the previous work that delineates the antibody levels and hesitancy among healthcare workers who received the COVID-19 vaccination. This finding is a small-scale study but provides some insight into outcomes. The methodology and outcomes were fine. However, some information needs to be addressed to clarify it.
Major concerns.
- Lines 82-84: Suggest clarifying the antibody targeting of each test to make it easy to understand to readers. Some readers who are not familiar with the serology will raise the question of why you determined two IgGs.
- Anti-nucleocapsid IgG (NCP)
- Anti-spike IgG (QuantiVac)
Furthermore, the immunologic outcomes, such as Figure 1, were mentioned as "anti-SARS-CoV-2 S IgG", but the Materials and Methods section did not mention what the anti-S is.
A1 – Thank you for your comment. We modified the Materials and Methods section and clarified the testing procedure and abbreviations.
‘Blood specimens were drawn by trained nurses at the DIDaCN in November 2022, i.e. at approximately one year after the first COVID-19 vaccine booster and two months after the second vaccine booster dose administration. Following collection, the blood samples were transported to the IBCH PAS for serological assays. In order to differentiate between Ab response induced by natural infection or vaccination we used immunoassays targeting two SARS-CoV-2 proteins: Spike (S) and Nucleocapsid (NCP). COVID-19 vaccines are based on S protein, so the increased anti-S Ab level is observed in both vaccinated and SARS-CoV-2 infected persons. On the contrary, anti-NCP Ab are produced only after SARS-CoV-2 infection and their elevated level is the indicator of previous natural exposure to SARS-CoV-2.
Analysis of anti-SARS-CoV-2 IgG Level
Serological testing for IgG Ab to the S protein and NCP protein was performed using a quantitative anti-SARS-CoV-2 QuantiVac IgG ELISA test (EuroImmun GmbH, Lübeck, Germany) and anti-SARS-CoV-2 NCP IgG ELISA test (EuroImmun GmbH, Lübeck, Germany), respectively. Presence of anti-S IgG Ab confirmed the response to either prior SARS-CoV-2 infection or vaccination, while presence of anti-NCP IgG Ab indicated only previous SARS-CoV-2 infection. Assays were carried out and interpreted as described in the producer's manuals. Results were calibrated into WHO international units (binding antibody unit, BAU/mL) using kit attached reference material and calibration curve. For each run new calibration curve was drawn and used for calculations. Analysis included HCWs with anti-SARS-CoV-2 IgG Ab measurement within 389±29 days after their first or 44±11 days after their second BNT162b2 booster vaccination. The time between the completing of primary homologous BNT162b2 vaccination series and anti-SARS-CoV-2 IgG Ab measurement was more than 18 months for HWCs who did not receive any BNT162b2 booster dose.’
Q2 – 2. Suggest adding the "conversion factor" of the QuantiVac assay to the Materials and Methods section because this assessment was not reported initially in the BAU/mL unit.
A2 – We used the QuantiVac assay with reference material from the National Institute for Biological Standards and Control, NIBSC, as written in the manufacturer’s instruction. Based on the results for the reference material, calibration curve was drawn and based on that we calculated the antibody titer in BAU/ml.
Q3 – Comments.
- Line 46: Suggest using "University of Oxford", a formal form of this university.
A3 – Thank you for your comment, we modified the name of the University.
Q4 – 2. Line 87: Suggests deleting "(EuroImmun GmbH) "because you were the first mentioned manufacturer of this test (NCP) in line 83.
A4 – Thank you for your comment. We modified the Materials and Methods section and indicated the manufacturer of the test when it was first mentioned.
Q5– 3. Line 91: Suggest moving "(EuroImmun GmbH)" to line 84 because you were the first mention of this test (QuantiVac) in line 84.
A5 – Thank you for your comment. We modified the Materials and Methods section and indicated the manufacturer of the test when it was first mentioned.
Q6 – 4. Line 88, 119 and elsewhere: Suggest using "BNT162b2" instead of "BNT162b2 mRNA COVID-19" to make it concise. The BNT162b2 is a very specific term that there is no need to mention further to make it lengthy.
A6 – Thank you for your comment, we deleted unnecessary information.
Q7 – 5. Lines 126-132: Suggest adding the median or mean interval day between the last dose and the blood draw to ensure the booster dose will increase immunity. Moreover, non-booster or first booster vaccination may cause the last dose for a long time, giving low immunity levels due to the immunity waning.
A7 – Thank you for your suggestion. We have modified manuscript accordingly.
‘Analysis included HCWs with anti-SARS-CoV-2 IgG Ab measurement within 389±29 days after their first or 44±11 days after their second BNT162b2 booster vaccination. The time between the completing of primary homologous BNT162b2 vaccination series and anti-SARS-CoV-2 IgG Ab measurement was more than 18 months for HWCs who did not receive any BNT162b2 booster dose.’
Q8– Typos.
- Line 45: "Astra Zeneca". Suggests using "AstraZeneca" (no spacing).
A8 –Thank you for your comment, we corrected the spelling.
Q9 – 2. Line 57: Suggests checking vaccine name "NVX-CoV2372". I think the FDA-approved vaccine is NVX-CoV2373.
A9 – Thank you for your comment, we corrected the spelling.
Q10 – 3. Suggest using "Pfizer—BioNTech" instead of "Pfizer/BioNTech" throughout the manuscript. The Pfizer—BioNTech is almost used form
A10 – Thank you for your comment, we corrected the spelling.
Reviewer 4 Report
Comments and Suggestions for Authors
The manuscript by Lorent et al., is interesting, however I do not feel the data presented is novel. The reluctance to receive COVID-19 booster doses for a wide range of reasons, some listed in the paper, has been well documented previously. In addition the observed decline in anti-NP, as time elapsed following natural infection, and the decline in anti-S following COVID-19 vaccination is not surprising and has also been well documented.
Therefore I do not believe the information presented in the manuscript merits publication
Author Response
Q1 – The manuscript by Lorent et al., is interesting, however I do not feel the data presented is novel. The reluctance to receive COVID-19 booster doses for a wide range of reasons, some listed in the paper, has been well documented previously. In addition the observed decline in anti-NP, as time elapsed following natural infection, and the decline in anti-S following COVID-19 vaccination is not surprising and has also been well documented. Therefore I do not believe the information presented in the manuscript merits publication
A1 – After reading the comment about lack of an innovative contribution, we rechecked the literature and could not find any examples of anyone having conducted a longitudinal study of COVID-19 vaccination among healthcare workers in Poland during the two years of the COVID-19 pandemic. We believe that our work is really innovative, because it deals with the problem of vaccination coverage and changing attitude of HCWs towards COVID-19 vaccination, and such information is lacking in Poland, as well as other European countries. As we showed in the introduction on the example of the Influenza virus, which has been a serious threat for many years, the problem of vaccine hesitancy is still unresolved, and despite many efforts, it is practically impossible to achieve the level of vaccination recommended by the WHO. It has been shown that the vaccination coverage depends on the context and country, and vaccination policy should be adjusted to the current situation, which is hampered by lack of data. We believe that our work will provide information that will improve the fight against SARS-CoV-2. More importantly, our data suggest that booster vaccination information campaign was not successful and the importance of booster doses is not well understood, even among HCWs. On the basis of the referees’ and editors comment we have added and supplement our data with additional information, and we believe that in the reviewed form the article can be published.